



# Atmospheric breakdown chemistry of the new green solvent 2,2,5,5-tetramethyloxolane via gas-phase reactions with OH and Cl radicals

Caterina Mapelli[1], Juliette V. Schleicher[1,2], Alex Hawtin[1], Conor D. Rankine[1,3], Fiona C. Whiting[1], Fergal Byrne[1,6], C. Rob McElroy[1], Claudiu Roman[4,5], Cecilia Arsene[4,5], Romeo I. Olariu[4,5], Iustinian G. Bejan[4,5], and Terry J. Dillon[1]

[1] Department of Chemistry, University of York, York, YO10 5DD, UK
[2] now at École Polytechnique Fédérale de Lausanne, 1015 Switzerland
[3] Department of Chemistry, Newcastle University, Newcastle upon Tyne, NE1 7RU, UK
[4] Faculty of Chemistry, "Alexandru Ioan Cuza" University of Iasi, Iasi, 11th Carol I, 700506, Romania
[5] Integrated Center of Environmental Science Studies in the North Eastern Region – CERNESIM, "Alexandru Ioan Cuza" University of Iasi, Iasi, 11th Carol I, 700506, Romania
[6] Department of Chemistry, Maynooth University, Maynooth, Co. Kildare, W23 F2H6, Ireland

*Correspondence to*: Terry J. Dillon (*terry.dillon@york.ac.uk*)

**Abstract.** The atmospheric chemistry of 2,2,5,5-tetramethyloxolane (TMO), a promising 'green' solvent replacement for toluene, was investigated in laboratory and computational experiments. Results from both absolute and relative rate studies demonstrated that the reaction OH + TMO (R1) proceeds with a rate coefficient $k_1(296\ K) = (3.1 \pm 0.4) \times 10^{-12}\ cm^3\ molecule^{-1}\ s^{-1}$, a factor of three smaller than predicted by recent structure activity relationships. Quantum chemical calculations (CBSQB3-G4) demonstrated that the reaction pathway via the lowest-energy transition state was characterised by a hydrogen-bonded pre-reaction complex, leading to thermodynamically less favoured products. Steric hindrance from the four methyl substituents in TMO prevent formation of such H-bonded complexes on the pathways to thermodynamically favoured products, a likely explanation for the anomalous slow rate of (R1). Further evidence for a complex mechanism was provided by $k_1(294 - 502\ K)$, characterised by a local minimum at around $T = 340\ K$. An estimated atmospheric lifetime of $\tau_1 \approx 3$ days was calculated for TMO, approximately 50% longer than toluene, indicating that any air pollution impacts from TMO emission would be less localised. Relative rate experiments were used to determine a rate coefficient, $k_2(296\ K) = (1.2 \pm 0.1) \times 10^{-10}\ cm^3\ molecule^{-1}\ s^{-1}$ for Cl + TMO (R2); together with the slow (R1) this may indicate an additional contribution to TMO removal in regions impacted by high levels of atmospheric chlorine. All results indicate that TMO is a less problematic volatile organic compound (VOC) than toluene.

## 1 Introduction

Aromatic solvents, unsustainably sourced from the petroleum industry, are harmful to health, often potentially carcinogenic, and environmentally hazardous (Jimenez-Gonzalez et al., 2011). Considerable research effort therefore has been directed



towards developing less harmful and hazardous new "green" solvents. Sustainably sourced from readily available, potentially renewable feedstock, 2,2,5,5-tetramethyloxolane (Fig. 1) also known as 2,2,5,5-tetramethyltetrahydrofuran, henceforth TMO, is a promising green solvent with unusual properties. Byrne et al. (2017) reported that the four methyl substituents alpha to the

ethereal oxygen lead to a low basicity and (by excluding alpha hydrogens) an inherent resistance to peroxide formation. TMO exhibits solvent behaviour similar to common hydrocarbons and has been proposed as a replacement compound for toluene (formally methylbenzene, $C_6H_5CH_3$) (Byrne et al., 2017). $C_6H_5CH_3$ is used at industrial sites and even in consumer products for a wide range of applications. Widespread in building work and coating products, as well as detergents and fragrances, $C_6H_5CH_3$ can be released in indoor and outdoor areas and its manufacturing reaches $10^6$ - $10^7$ tonnes per year in the European

Economic Area (Echa, 2022). Both $C_6H_5CH_3$ and TMO are examples of volatile organic compounds (VOC), i.e. organic liquids with significant vapour pressures at ambient temperature. The atmospheric oxidation of VOC is governed by complex mechanisms and proceeds via a host of partially-oxidised intermediates (Atkinson, 1986; Atkinson and Arey, 2003; Jenkin et al., 1997; Saunders et al., 2003). The crucial, often rate-determining, step is the first attack on the VOC from oxidants such as ozone ($O_3$), gas-phase free-radicals, or directly via photolysis. For $C_6H_5CH_3$, the principal breakdown route is via bimolecular

reaction (R3) with the hydroxyl radical (OH).

OH + $C_6H_5CH_3$ → (products)                                                                                                          (R3)

IUPAC have evaluated the kinetic literature for (R3) and recommend a rate coefficient, $k_3$(298 K) = $5.6\times10^{-12}$ cm$^3$ molecule$^{-1}$ s$^{-1}$ with an associated uncertainty $\Delta\log k_3$(298 K) = 0.10 or approximately 25% (Mellouki et al., 2021). The removal rate of $C_6H_5CH_3$ from the troposphere is therefore reasonably well established, with a photochemical lifetime of around two days.

The subsequent oxidative breakdown of aromatics such as $C_6H_5CH_3$ in air is known to yield rapidly large quantities of harmful ozone ($O_3$), formaldehyde (HCHO), and particulates (Hansen et al., 1975; Saunders et al., 2003; Bloss et al., 2005). Waste and / or inefficient use of aromatic VOC is therefore an established source of harmful gaseous emissions to the atmosphere. Overall mortality from poor air quality has been estimated as a cause of over 400,000 annual deaths in Europe alone (EEA Report 9/2020 (Eea, 2020), (Lelieveld et al., 2019)). By contrast with established pollutants, little to nothing is known of the

atmospheric chemistry of TMO. In the absence of a near-UV chromophore (Christianson et al., 2021) and any C=C double bonds, the principal route for breakdown of TMO in the troposphere is likely via reaction (R1) with OH (Fig. 1). In air heavily impacted by atmospheric chlorine (Ariya et al., 1999; Atkinson and Aschmann, 1985; Thornton et al., 2010), TMO loss may be augmented by reaction with Cl atoms (R2).

OH + TMO → (products)                                                                                                                (R1)
Cl + TMO → (products)                                                                                                                (R2)

To date there appears to be no gas-phase kinetic data on (R1-R2) from which to calculate the reactivity and properties such as the atmospheric lifetime of TMO. In the absence of such data, even a cursory assessment of the environmental impact of
replacing $C_6H_5CH_3$ with TMO is not feasible. Accordingly, the objectives of this work were to use lab-based experiments to



determine accurate values of $k_1$ and $k_2$ and to interpret these results to make informed predictions as to the subsequent breakdown chemistry of TMO (Fig. 1).

## 2 Experimental

Laboratory-based experiments were carried out using two well-established kinetic techniques: pulsed laser photolysis (PLP) generation of OH coupled to direct detection by laser-induced fluorescence (LIF) for absolute determinations of $k_1$(294 – 502 K) as detailed in section 2.1; and a smog chamber equipped with long path Fourier transform infra-red spectroscopy (FTIR) for detection of TMO and suitable reference VOC for relative rate determinations of $k_1$(296 K) and $k_2$(296 K) as detailed in section 2.2. Details of the quantum chemical calculations (QCC) used to investigate TMO and OH – TMO complexes are

provided in section 2.3.

### 2.1 Direct PLP-LIF determinations of $k_1(T)$

Absolute laser-based kinetic experiments were carried out at University of York. Photochemistry took place in an approximately spherical Pyrex reactor vessel of volume $\approx 400$ cm$^3$ equipped with laser side-arms and gas inlets (ca. 30 cm

upstream of the photolysis region; see Fig. 2). All were wrapped with heating tape and insulated with glass-fibre tape; a K-type thermocouple situated between the walls of the reactor and the insulation tape was coupled to feedback electronics to allow for temperature control. Temperature in the reactor itself was monitored with a second K-type thermocouple that could be readily translated in and out of the photolysis region. Pressure was monitored by two calibrated capacitance manometers (MKS, 10 and 1000 Torr). Gas flow rates were regulated via four calibrated mass flow controllers (MFC); gases were pre-

mixed upstream of the reaction cell. Absolute determinations of $k_1(T)$ rely critically on knowledge of TMO concentration, here calculated to an estimated precision of ± 15% from manometric measurements.

The fourth harmonic output ($E = 100$ mJ cm$^{-2}$ / pulse) from a Nd:YAG laser (Quantel, Q-Smart) operating at 10 Hz was directed into the reactor via a quartz Brewster window and used to generate OH via 266 nm PLP (R4 or R5) of one of two suitable radical precursors.


$$H_2O_2 + h\upsilon \rightarrow 2OH \tag{R4}$$

$$(CH_3)_3COOH + h\upsilon \rightarrow (CH_3)_3CO + OH \tag{R5}$$

Kinetic experiments conducted in the absence of TMO and manometric estimates indicated that concentrations of either $H_2O_2$

or $(CH_3)_3COOH$ were low ($< 10^{14}$ molecule cm$^{-3}$) in all experiments. Under these conditions, an estimated [OH] $\approx 10^{11}$ molecule cm$^{-3}$ was generated (R4 or R5). As such, pseudo-first-order conditions of [TMO] >> [OH] applied throughout, save in experiments conducted with [TMO] = 0.





The 282 nm output from a frequency-doubled dye laser (Radiant dyes using Rhodamine-6G; pumped at 532 nm by a YAG, Continuum) was directed through a second quartz Brewster window, co-linear but counter-propagating to the PLP laser (see Fig. 2). This tuneable laser light was used to pump the $Q_{11}$ transition of $A^2\Sigma^+(v = 1) \leftarrow X^2\Pi(v = 0)$ at 281.997 nm for direct, off-resonant LIF detection of OH. Perpendicular to the laser beams, a pair of biconvex lenses was used to collect the 308 nm fluorescence from OH, direct it through a 309 nm interference filter, and onto a photomultiplier (PMT, Hamamatsu). The PMT output was passed through a fast pre-amplifier to an oscilloscope (Picoscope 6000) for collection, digitisation, and analysis.

Chemicals: $N_2$ > 99.9999% was obtained directly from $N_2$(l) boil-off; $O_2$ (99.995%, BOC) was used as supplied; $H_2O_2$ (JT Baker, 60% in $H_2O$) was prepped to an estimated (vapour pressure) mixing ratio of > 90% by continuous flow of $N_2$ through the liquid to remove the more volatile $H_2O$ component then supplied via a bubbler maintained at $T = 273$ K, situated downstream from a MFC and thus operating at close to reactor pressure; $(CH_3)_3COOH$ (70%, SigmaAldrich, Luperoxr TBH70X) was supplied from a Pyrex storage bulb (mixing ratio ≈ 2% in $N_2$); TMO (> 97%, synthesised within the Green Chemistry Centre of Excellence, University of York) and $CH_3OH$ (99.9%, SigmaAldrich) were both subject to repeated freeze-pump-thaw cycles at $T = 77$ K prior to dilution with $N_2$ (mixing ratios ≈ 1% for TMO, ≈ 3% for $CH_3OH$) in 12 dm$^3$ Pyrex bulbs for storage and supply.

## 2.2 Smog chamber studies of (R1) and (R2)

Relative rate experiments were conducted in the 760 dm$^3$ ESC-Q-UAIC environmental simulation chamber at "Alexandru Ioan Cuza" University of Iasi, Romania (Roman et al., 2022). The quartz chamber was equipped with inlet ports, sampling lines, 2 sets of UV lamps for photolysis at 365 nm (generation of OH or Cl-atoms) and at 254 nm (generation of OH), and FTIR instrumentation for observation of reference VOC, TMO, and TMO oxidation products. All experiments were conducted at $p = (1000 \pm 10)$ mbar (synthetic air) and $T = (296 \pm 2)$ K. In preliminary experiments, wall deposition and photolysis rates (360 nm and 254 nm) were measured to ensure the validity of the experiment and to correct raw (R1) or (R2) kinetic data as necessary. Spectral lines were tested for proportionality to the concentration of TMO and of reference compounds and for available subtraction spectral features of TMO and reference compounds. Two distinct OH generation methods were used. $CH_3ONO$ was injected into the chamber for photolysis at 365 nm (R6). The presence of an excess of NO ensured rapid conversion (R7 – R8) to OH.

$$CH_3ONO + h\nu \rightarrow CH_3O + NO \tag{R6}$$
$$CH_3O + O_2 \rightarrow HCHO + HO_2 \tag{R7}$$
$$HO_2 + NO \rightarrow OH + NO_2 \tag{R8}$$





Alternatively, OH was generated directly via 254 nm photolysis of $H_2O_2$ (R4). For all $k_2$ determinations, atomic chlorine was generated by the photolysis of $Cl_2$ at 365 nm (R9).

$$Cl_2 + h\nu \rightarrow 2Cl \hspace{6cm} \text{(R9)}$$

Choice of reference compound for kinetic experiments was informed by two factors. First, regarding FTIR spectra,
experimental and existing library data were used to ensure that reference compound spectra contained strong features that did not overlap with those of TMO (Fig. 3). Second, reference compound $k$(296 K) data needed to be well-characterised and available in the literature (see Table 2); these values were preferably of a similar magnitude to measured (PLP-LIF) or predicted $k_1$ and $k_2$ values. According to these principles the selected reference compounds for the reaction with OH were dimethylether ($CH_3OCH_3$) and cyclohexane ($c$-$C_6H_{12}$), as represented in Table 2 (R10, R11). Since no absolute method was available to study
Cl + TMO (R2), a third reference compound (propene, $C_3H_6$) was selected for these relative rate experiments (Table 2, R12 - R14). All reference VOC, TMO, and TMO breakdown products were monitored by FTIR (Fig. 3). Spectra were recorded every 2 minutes by combining 120 scans for a spectrum, with approximately 15 such spectra at a resolution of $1 cm^{-1}$ completing each experiment. Conversion of TMO and reference compounds through the reaction with OH and Cl radicals has been achieved at least to 50%.


**Table 2. Literature kinetic data for reference compounds used in determinations of $k_1$(296 K) and $k_2$(296 K)**

| Reaction | Reference Compound | $k^a$ | Method | References |
|---|---|---|---|---|
| R10 | $CH_3OCH_3$ + OH | 2.86 | Relative rate (GC) | (Demore and Bayes, 1999) |
| R11 | $c$-$C_6H_{12}$ + OH | 6.97 | Review | (Atkinson, 1986) |
| R12 | $CH_3OCH_3$ + Cl | 173 | Relative rate (GC) | (Giri and Roscoe, 2010) |
| R13 | $c$-$C_6H_{12}$ + Cl | 330 | Relative rate (GC-MS) | (Anderson et al., 2007) |
| R14 | $C_3H_6$ + Cl | 223 | Relative rate (FTIR) | (Ceacero-Vega et al., 2009) |
| **Notes: a** = $k$ values in $10^{-12}$ $cm^3$ molecule$^{-1}$ s$^{-1}$ are for conditions of $p$ = 1000 ± 10 mbar and $T$ = 296 ± 2 K | | | | |

Chemicals: Liquid samples of the following were supplied to the reactor by direct injection through a septum: $H_2O_2$ (60% in $H_2O$); TMO (> 97%, synthesised within the Green Chemistry Centre of Excellence, University of York); $CH_3ONO$ was
prepared in CERNESIM laboratory using an adapted method from Taylor et al. (1980); $CH_3OCH_3$ >99.9% (suitable for GC analysis, Sigma-Aldrich), $c$-$C_6H_{12}$ 99.5% (anhydrous, Sigma-Aldrich), and $C_3H_6$ >99.9% (Sigma-Aldrich) were used as supplied.





### 2.3 Quantum chemical calculations (QCC)

QCC were carried out using Gaussian 09 rev. D01 (Frisch et al., 2016) at the complete basis set QB3 (CBS-QB3;Montgomery et al. (1999)) and Gaussian-4 (G4;Curtiss et al. (2007)) levels of theory. An SCF convergence criterion of $1.0 \times 10^{-6}$ a.u. was used in all calculations; convergence criteria of $1.0 \times 10^{-6}$, $3.0 \times 10^{-4}$, and $1.2 \times 10^{-3}$ a.u. were used for the energy change, RMS gradient, and RMS displacement, respectively, in all geometry optimisations. All geometries at which properties were evaluated at the CBS-QB3 level were optimised using density functional theory (DFT) at the B3LYP ((Becke, 1993);(Lee et al., 1988)) level with the 6-311G($2d,d,p$) (CBSB7) basis set; all geometries at which properties were evaluated at the G4 level were optimised using DFT at the B3LYP level with the 6-31G($2df,p$) (GTBAS3) basis set. The proper convergence of all geometry optimisations to the stationary point of interest was verified via vibrational frequency inspection- one imaginary frequency for first-order saddle points (also called transition states, TS) and no imaginary frequency for local minima.

### 3 Results and discussion

Results from two experimental studies and from quantum chemical calculations are described below. Section 3.1 details determinations of $k_1(294 – 490$ K) in direct, absolute laser-based experiments. In section 3.2 the results from complementary relative-rate experiments to determine $k_1(296$ K) and $k_2(296$ K) are presented. Section 3.3 contains results from QCC to explore the structure and reactivity of TMO and its interactions with OH whilst the discussion section 3.4 attempts to rationalise these results and to predict products. Uncertainties (±) quoted throughout this work are two sigma statistical only, derived from regression analysis, unless specifically stated otherwise.

### 3.1 PLP-LIF determinations of $k_1(T)$

PLP-LIF studies were carried under pseudo-first order conditions of [TMO] >> [OH] such that OH LIF time profiles, $S(t)$, were described by a monoexponential decay, expression Eq. (1):

$$S(t) = S_0 \exp(-Bt)$$ Eq. (1)

The parameter $S_0$ describes (in arbitrary units) the LIF signal at $t = 0$ and is proportional to the initial [OH] produced by the laser pulse in (R4) or (R5). The parameter $B$ is the pseudo-first-order rate coefficient for OH decay (and includes components from both reactive and transport losses). Figure 4 displays typical OH decay profiles recorded in the presence of three different excess [TMO], whilst other conditions of $P = 100$ mbar (N$_2$), $T = 296$ K, and [H$_2$O$_2$] $\approx 1 \times 10^{14}$ molecule cm$^{-3}$ were unchanged. These and other similar OH LIF profiles were typically exponential over at least an order of magnitude and were fit with Eq. (1) to yield values of $B$ with a high degree of precision (standard errors were generally less than ± 5%).





Systematic errors from unwanted radical chemistry were considered unlikely. Use of low [OH] ensured that losses of OH by reaction with itself or with (R1) or (R5) products were minimal. Further, cyclic / aliphatic ethers do not absorb appreciably at wavelength longer than 200 nm (Christianson et al., 2021), so preventing generation of organic radical fragments in the laser
flash. Nonetheless, two series of experiments were performed to test for the impact of unanticipated secondary chemistry. First, no systematic change in value of parameter $B$ was obtained upon changing the photolysis laser fluence by a factor of three (via modifications to the Q-switch delay). Second, a series of "back-to-back" experiments were performed where the bath gas was alternated between $N_2$ and air. Once again, no systematic changes in parameter $B$ were observed ($B_{air}$ / $B_{N2}$ = 1.01 $\pm$ 0.05) from over thirty such paired experiments conducted at pressures of between 20 and 100 Torr. Taken together these
observations indicate that the influence of any secondary chemistry was negligible.

Figure 5 displays plots of parameter $B$ vs. [TMO] at three temperatures; each were fit with Eq. (2) to obtain values of $k_1(T)$,

$$B = k_1[\text{TMO}] + k_{loss} \qquad \text{Eq. (2)}$$

where the term $k_{loss}$ ($s^{-1}$) accounts for other losses of OH, here dominated by reaction (R15) with the photolysis precursor $H_2O_2$
with some small contribution from diffusion and flow out of the reaction zone.

$$\text{OH} + \text{H}_2\text{O}_2 \rightarrow \text{H}_2\text{O} + \text{HO}_2 \qquad \text{(R15)}$$

The data display good linearity, with the slopes identified as $k_1(T)$ at three different temperatures and intercept values (around a few hundred $s^{-1}$) in line with the predicted rate for (R15) with an estimated $[\text{H}_2\text{O}_2.] = 10^{14}$ molecule $cm^{-3}$. A mean of four values obtained at around room temperature yielded $k_1(296 \text{ K}) = (3.07 \pm 0.04) \times 10^{-12} \text{ cm}^3 \text{ molecule}^{-1} \text{ s}^{-1}$, independent of the
bath gas pressure or identity ($N_2$ or air). To the best of our knowledge, these results represent the first such kinetic data for (R1). Experimental conditions and results of all PLP-LIF determinations of $k_1$ are listed in Table 3. Note that these absolute determinations of $k_1$ were critically dependent on reliable determination of [TMO] via manometric measurements. Every effort was made to calibrate MFC and pressure gauges, and to conduct experiments using a variety of conditions of pressure, flow rate, and different TMO supply bulbs. Many of the experiments to determine $k_1(T)$ were conducted "back-to-back" with the
well-characterised reaction (R16) which is known to proceed with a rate coefficient of approximately similar magnitude.

$$\text{OH} + \text{CH}_3\text{OH} \rightarrow \text{(products)} \qquad \text{(R16)}$$

**Table 3** lists full results for absolute kinetic experiments to determine $k_1(T)$ and $k_{16}(T)$ in this work.

**Table 3: Absolute $k_1(T)$ for OH + TMO and $k_{16}(T)$ for OH + CH$_3$OH determined via PLP-LIF in this work.**

| OH + VOC | $T$ / K | $p$ / Torr | OH precursor | $n$ [a] | [VOC] [b] | $k(T)$ [c] |
|---|---|---|---|---|---|---|
| TMO (R1) | 298 | 30 | (CH$_3$)$_3$COOH, (R5) | 9 | 1-28 | 2.9 $\pm$ 0.2 |
| TMO (R1) | 297 | 60 | (CH$_3$)$_3$COOH, (R5) | 11 | 1-28 | 3.1 $\pm$ 0.6 |
| TMO (R1) | 296 | 60 | H$_2$O$_2$, (R4) | 16 | 1-37 | 3.1 $\pm$ 0.2 |



| TMO (R1) | 297 | 59 | $H_2O_2$, (R4) | 10 | 2-15 | $3.2 \pm 0.2$ |
|---|---|---|---|---|---|---|
| TMO (R1) | 314 | 58 | $H_2O_2$, (R4) | 8 | 2 – 18 | $2.75 \pm 0.3$ |
| TMO (R1) | 336 | 34 | $(CH_3)_3COOH$, (R5) | 10 | 1 – 30 | $2.7 \pm 0.3$ |
| TMO (R1) | 344 | 58 | $H_2O_2$, (R4) | 8 | 2 – 18 | $2.7 \pm 0.3$ |
| TMO (R1) | 383 | 58 | $H_2O_2$, (R4) | 8 | 2 - 25 | $3.0 \pm 0.1$ |
| TMO (R1) | 421 | 60 | $H_2O_2$, (R4) | 10 | 1 - 14 | $3.4 \pm 0.3$ |
| TMO (R1) | 441 | 60 | $H_2O_2$, (R4) | 8 | 1 - 12 | $3.6 \pm 0.3$ |
| TMO (R1) | 464 | 29 | $H_2O_2$, (R4) | 8 | 1 - 15 | $4.7 \pm 0.5$ |
| TMO (R1) | 502 | 29 | $H_2O_2$, (R4) | 13 | 1 - 14 | $5.9 \pm 0.3$ |
| $CH_3OH$ (R16) | 299 | 30* | $(CH_3)_3COOH$, (R5) | 10 | 3.5 - 97.1 | $0.84 \pm 0.03$ |
| $CH_3OH$ (R16) | 299 | 30 | $(CH_3)_3COOH$, (R5) | 11 | 3.5 - 97.1 | $0.87 \pm 0.07$ |
| $CH_3OH$ (R16) | 346 | 62 | $H_2O_2$, (R4) | 14 | 3.2 - 85.4 | $1.44 \pm 0.08$ |
| $CH_3OH$ (R16) | 400 | 38 | $H_2O_2$, (R4) | 8 | 1.6 - 44.0 | $1.5 \pm 0.2$ |
| $CH_3OH$ (R16) | 441 | 60 | $H_2O_2$, (R4) | 12 | 3.4 - 34.0 | $1.8 \pm 0.1$ |
| $CH_3OH$ (R16) | 471 | 29 | $H_2O_2$, (R4) | 9 | 4.0 - 38.0 | $1.8 \pm 0.1$ |
| $CH_3OH$ (R16) | 501 | 29 | $H_2O_2$, (R4) | 7 | 3.4 - 40.0 | $2.4 \pm 0.2$ |

**Notes**: [a] $n$ is the number of different [VOC] used (not including [VOC] = 0); [b] = range of [VOC] in units of $10^{13}$ molecule cm$^{-3}$ ; [c] = $k(T)$ in units of $10^{-12}$ cm$^3$ molecule$^{-1}$ s$^{-1}$; bath gas M = $N_2$ unless denoted * where M = air.


Figure 6 presents the results from these PLP-LIF experiments in Arrhenius format. Results for (R16) were in reasonable agreement with previous determinations (Wallington et al., 1988; Hess and Tully, 1989; Jiménez et al., 2003; Dillon et al., 2005) and with a three-parameter expression, $k_{11}(210 – 866$ K$) = 6.38 \times 10^{-18} T^2 \exp(144 / T)$ cm$^3$ molecule$^{-1}$ s$^{-1}$, recommended by IUPAC (Atkinson et al., 2006) (see Fig. 6). This satisfactory agreement lends some confidence to the assessment of
uncertainties in $k_1(T)$ for this work. Over the limited range of temperatures explored in this work, the results for (R16) appear to conform closely to the Arrhenius equation, with the smallest values of $k_{16}(T)$ found at the lowest temperature. The contrast with the $k_1(294 – 490$ K$)$ data was stark, where a local minimum was observed at around $T = 340$ K; values of $k_1$ increased to both lower and to higher temperatures. Potential explanations for this non-Arrhenius behaviour are explored in section 3.3





below, where QCC were used to explore OH – TMO interactions. A comparison with results for similarly oxygenated VOC +
OH reactions is presented in section 3.4.

### 3.2 Relative rate determinations of $k_1$(296 K) and $k_2$(296 K)

Figure 3 displays FTIR spectra of TMO recorded at the CERNESIM atmospheric simulation chamber (Roman et al., 2022).
Qualitatively, the spectra compare well to predictions (Frisch et al., 2016), thus allowing deployment of pre-planned reference
compounds (Table 2) for relative rate experiments. Experiments conducted in the absence of radical precursors demonstrated
that neither TMO nor the various reference VOC (Table 2) were significantly impacted by wall losses or photolytic removal,
indicating that corrections to subsequent kinetic data were unnecessary. FTIR peak intensities were directly proportional to
species concentrations and were used to calculate the logarithmic depletion for TMO and for the reference compound. Results
from studies using $c$-$C_6H_{12}$ and $CH_3OCH_3$ as reference compounds to determine $k_1$(296 K) are displayed in Fig. 7. According
to Eq. (3), the slope from this plot should be a straight line with almost "zero" intercept and be identified with the relative rate
$k_1 / k_{ref}$.

$$ln\frac{[TMO]_{t_0}}{[TMO]_t} = \frac{k_1}{k_{ref}} ln\frac{[reference]_{t_0}}{[reference]_t} \qquad \text{Eq. (3)}$$

The results obtained from these and a similar series of experiments are presented in Table 4. The spread of values seems
reasonable given both statistical uncertainties and the (approximately 20%) systematic uncertainties in the literature reference
$k$-values. A weighted mean value from these four relative rate determinations was $k_{1,RR}$(296 K) $= (3.07 \pm 0.05) \times 10^{-12}$ cm$^3$
molecule$^{-1}$ s$^{-1}$.

**Table 4: results from relative rate experiments to determine $k_1$(296 K) for OH + TMO from this work**

| OH precursor | Ref. VOC | $\lambda$ / nm | [Ref]$_0$ [a] | [TMO]$_0$ [a] | $k_1 / k_{refOH}$ [b] | $k_1$(296 K) [c] |
|---|---|---|---|---|---|---|
| $CH_3ONO$ (R6 – R8) | $c$-$C_6H_{12}$ (R11) | 365 | 3.30 | 3.28 | $0.435 \pm 0.009$ | $3.03 \pm 0.07$ |
| $CH_3ONO$ (R6 – R8) | $c$-$C_6H_{12}$ (R11) | 365 | 2.35 | 1.48 | $0.44 \pm 0.01$ | $3.10 \pm 0.08$ |
| $CH_3ONO$ (R6 – R8) | $CH_3OCH_3$ (R10) | 365 | 3.30 | 2.84 | $1.14 \pm 0.07$ | $3.3 \pm 0.2$ |
| $H_2O_2$ (R9) | $CH_3OCH_3$ (R10) | 254 | 3.30 | 2.84 | $1.01 \pm 0.05$ | $2.9 \pm 0.2$ |
| **Notes**: all experiments were conducted at $p$ = 1 bar (air) and $T$ = 296 K ; [a] = initial [Ref] and [TMO] values calculated from the injected amount (in mass units) in $10^{13}$ molecule cm$^{-3}$; ; [b] = see Table 2 for values of $k_{12}$(298 K) and $k_{13}$(298 K) used here as $k_{ref}$ ; [c] = $k_1$(296 K) in units of $10^{-12}$ cm$^3$ molecule$^{-1}$ s$^{-1}$. | | | | | | |


A similar series of experiments were conducted to determine $k_2$(296 K) for Cl + TMO (R2). Details of reference compounds
used were given in Table 2, exemplary results presented in Figure 8, and the full dataset summarized in Table 5. The weighted





mean value obtained was $k_{2,RR}(296 \text{ K}) = (1.2 \pm 0.1) \times 10^{-10}$ cm³ molecule⁻¹ s⁻¹, with $k_2(296 \text{ K}) = (1.2 \pm 0.3) \times 10^{-10}$ cm³ molecule⁻¹ s⁻¹ representing a more realistic overall uncertainty estimate. This result is, to the best of our knowledge, the first

such reported value for (R2).

**Table 5: results from relative rate experiments to determine $k_2(296 \text{ K})$ for Cl + TMO from this work**

| Ref. VOC | $[\text{Ref}]_0$ [a] | $[\text{TMO}]_0$ [a] | $k_2 / k_{\text{refCl}}$ [b] | $k_2(296 \text{ K})$ [c] |
|---|---|---|---|---|
| $CH_3OCH_3$ (R12) | 3.30 | 3.71 | $0.76 \pm 0.01$ | $1.31 \pm 0.02$ |
| $CH_3OCH_3$ (R12) | 6.60 | 4.90 | $0.68 \pm 0.01$ | $1.18 \pm 0.02$ |
| $c$-$C_6H_{12}$ (R13) | 1.47 | 5.32 | $0.37 \pm 0.01$ | $1.22 \pm 0.08$ |
| $c$-$C_6H_{12}$ (R13) | 2.20 | 4.90 | $0.349 \pm 0.005$ | $1.15 \pm 0.02$ |
| $C_2H_6$ (R14) | 6.57 | 5.32 | $0.55 \pm 0.01$ | $1.23 \pm 0.03$ |
| $C_2H_6$ (R14) | 6.57 | 4.90 | $0.425 \pm 0.01$ | $0.95 \pm 0.03$ |

**Notes**: all experiments were conducted at $p = 1000$ mbar (air) and $T = 296$ K using 365 nm $Cl_2$ photolysis to generate Cl-atoms; [a] = initial $[\text{Ref}]_0$ and $[\text{TMO}]_0$ calculated from the injected amount (in mass units) in $10^{13}$ molecule cm⁻³; [b] = see Table 2 for values of $k_{\text{ref,Cl}}$ used here ; [c] = $k_2(296 \text{ K})$ from this work reported in $10^{-10}$ cm³ molecule⁻¹ s⁻¹.

### 3.3 Quantum Chemical Calculations on (R1)

QCC were carried out at the CBS-QB3 (Montgomery et al., 1999) and G4 (Curtiss et al., 2007) model chemical levels of theory (Section 2.3) to investigate the small $k_1(296 \text{ K})$ and complex $k_1(T)$ determined in laboratory studies, explore the hydrogen abstraction pathways (R1a) and (R1b) set out in Figure 1 and thus predict the products of (R1).

Acknowledging the $C_2$ symmetry of TMO and assuming pseudo-equivalence of the (strictly non-symmetric) methyl substituents, it was possible to characterise a total of five reaction channels. Three were associated with generation of (R1b)

products, corresponding to abstraction of each of the three unique hydrogen atoms at the $\beta$ position on a methyl (-$CH_3$) substituent; two were associated with (R1a), corresponding to abstraction of each of the two unique hydrogen atoms of the $\beta$ methylene (-$CH_2$-) on the aliphatic ring. For each reaction channel, transition states (TS) for hydrogen abstraction were located and characterised ($TS_{1-3}^{R1b}$ and $TS_{4-5}^{R1a}$ depicted in Figure 9), together with corresponding pre- (OH + TMO) and post- [$H_2O$ + TMO(−H)] reaction complexes. Cartesian coordinates for all key geometries are given in Tables S1-20 (CBS-QB3) and Tables

S21-40 (G4), while electronic and free energies at the CBS-QB3 and G4 levels of theory are summarised in Tables S41 and S42, respectively. The free energy profiles of the reaction channels are presented in Figure 10, while a summary of the relative free energies ($\Delta G^{\ddagger}_{298K}$), mass-weighted distances (MWDs) quantifying the separation in Cartesian space between the TS and pre-reaction complexes, and key structural parameters ($r$O–H, $r$O···H, and $a$H–O···H) are tabulated for each of the TS in Table 6.



The energetic barrier $\Delta G^{\ddagger}_{298K}$ along the reaction coordinate was similar for hydrogen abstraction pathways (R1a) and (R1b); a tight spread of $\Delta G^{\ddagger}_{298K}$ values [31.0 → 38.3 kJ mol$^{-1}$ (CBS-QB3); 26.9 → 32.6 kJ mol$^{-1}$ (G4)] was calculated by comparison with the spread of $\Delta G_{298K}$ values for either the pre- [−1.0 → 29.6 kJ mol$^{-1}$ (CBS-QB3); −8.5 → 20.6 kJ mol$^{-1}$ (G4)] or post-reaction [–47.6 → −63.7 kJ mol$^{-1}$ (CBS-QB3); –54.8 → −72.0 kJ mol$^{-1}$ (G4)] complexes. These QCC indicate a weak preference for hydrogen abstraction proceeding via stabilised, hydrogen-bonded pre-complexes and TS (TS$_1^{R1b}$ and TS$_3^{R1b}$)

over 'direct' hydrogen abstraction (TS$_2^{R1b}$, TS$_4^{R1a}$, and TS$_5^{R1a}$) at the CBS-QB3 level of theory (ca. 4 – 5 kJ mol$^{-1}$) that is less pronounced at the G4 level of theory (ca. 1 – 2 kJ mol$^{-1}$). Thus, no clear indication of the dominant reaction channel under kinetic control was obtained from inspection of $\Delta G^{\ddagger}_{298K}$ alone. However, $\Delta G^{\ddagger}_{298K}$ is likely not decisive for the outcome of the reaction of OH + TMO; the free energy surface topography along the reaction coordinate associated with the approach of OH and onwards to pre-complex formation is likely to influence the outcome to a large extent. The loosely-bound pre-reaction

complexes associated with TS$_2^{R1b}$, TS$_4^{R1a}$, and TS$_5^{R1a}$ are located in a less-accessible, higher-energy part of the free energy surface [ca. 25 – 30 kJ mol$^{-1}$ (CBS-QB3); ca. 15 – 20 kJ mol$^{-1}$ (G4)], while the hydrogen-bound pre-reaction complex through which the reaction channel bifurcates towards TS$_1^{R1b}$ and TS$_3^{R1b}$ is submerged [ca. −1.0 kJ mol$^{-1}$ (CBS-QB3); ca. -8.5 kJ mol$^{-1}$ (G4)] relative to reactants. The intimation is that the hydrogen-bound pre-reaction complex acts as a funnel on the free energy surface to bias the reaction towards preferential production of the kinetic (via R1b), rather than the thermodynamic (via R1a),

radical products (Fig. 1).

**Table 6: summary of relative TS free energies ($\Delta G^{\ddagger}_{298K}$),[a] MWDs ($d$)[b] between TS and pre-reaction complexes, and key structural parameters ($r$O–H,[c] $r$O···H,[d] and $a$H–O···H) for TS$_{1-3}^{R1b}$ and TS$_{4-5}^{R1a}$ at the CBS-QB3 and G4 levels of theory**

| TS | CBS-QB3 | | | | | G4 | | | | |
|---|---|---|---|---|---|---|---|---|---|---|
| | $\Delta G^{\ddagger}_{298K}$ | $d$ | $r$O–H | $r$O···H | $a$H–O···H | $\Delta G^{\ddagger}_{298K}$ | $d$ | $r$O–H | $r$O···H | $a$H–O···H |
| TS$_1^{R1b}$ | 32.1 | 11.58 | 0.972 | 1.320 | 94.9 | 28.7 | 12.48 | 0.972 | 1.314 | 94.8 |
| TS$_2^{R1b}$ | 38.3 | 3.02 | 0.971 | 1.380 | 97.7 | 32.6 | 2.73 | 0.971 | 1.375 | 97.7 |
| TS$_3^{R1b}$ | 31.0 | 5.01 | 0.974 | 1.314 | 95.2 | 26.9 | 4.92 | 0.974 | 1.307 | 95.0 |
| TS$_4^{R1a}$ | 36.2 | 5.39 | 0.972 | 1.411 | 97.4 | 28.5 | 5.34 | 0.972 | 1.399 | 97.5 |
| TS$_5^{R1a}$ | 37.1 | 0.84 | 0.972 | 1.531 | 96.3 | 29.2 | 0.81 | 0.972 | 1.515 | 96.2 |

**Notes**: [a] = tabulated in kJ mol$^{-1}$ at $p$ = 1 000 mbar (air) and $T$ = 298 K relative to the reactants (OH + TMO) at infinite separation; [b] = tabulated in Å Da$^{-\frac{1}{2}}$; [c] = the O–H internuclear distance in the OH fragment, tabulated in Å; [d] = the O···H internuclear distance between the OH and TMO fragments, tabulated in Å.

The formation of hydrogen-bound pre-reaction complexes and TS in the R1a channels is precluded by strong steric interactions with methyl substituents on TMO (cf. hydrogen abstraction from the $\beta$ position on the aliphatic ring in the unsubstituted analogue, oxolane (tetrahydrofuran), where a hydrogen-bound pre-reaction complex and TS is able to lower effectively the





energetic barrier to the (R1a) equivalent products) and, although their formation is possible in the (R1b) channels, the conferred stabilisation is somewhat offset by a combination of i) weak steric interactions with, and ii) charge screening by, the methyl

substituents.

**3.4 Discussion**

The lab results from this work of $k_1(296 \pm 2$ K) = $(3.07 \pm 0.04) \times 10^{-12}$ cm$^3$ molecule$^{-1}$ s$^{-1}$ (from direct, absolute PLP-LIF) and $k_1(296$ K) = $(3.07 \pm 0.05) \times 10^{-12}$ cm$^3$ molecule$^{-1}$ s$^{-1}$ (relative rate), were in good agreement with one another, especially when considering systematic uncertainties. That these kinetic methods are based upon different critical assumptions (absolute

knowledge of [TMO] for PLP-LIF, reference rate coefficients for relative-rate) and have complementary strengths and weaknesses, lends considerable confidence to these results. Nonetheless, taking into account systematic uncertainties inherent in these experiments (in [TMO] for the PLP-LIF studies; the sizable uncertainties in reference rate coefficients in the relative rate study) we consider it appropriate to quote a more conservative overall $k_1(296$ K) = $(3.1 \pm 0.4) \times 10^{-12}$ cm$^3$ molecule$^{-1}$ s$^{-1}$ from this work. We therefore conclude that, at ambient temperature, OH reacts with TMO considerably more slowly than with

other oxolanes, e.g.: oxolane (tetrahydrofuran) $k = 1.7 \times 10^{-11}$ cm$^3$ molecule$^{-1}$ s$^{-1}$ (Moriarty et al., 2003); 2-methyloxolane, $k = 2.65 \times 10^{-11}$ cm$^3$ molecule$^{-1}$ s$^{-1}$ (Illes et al., 2016); and 2,5-dimethyloxolane for which only data for the reaction with isotopically-labelled OD is available and where $k = 4.6 \times 10^{-11}$ cm$^3$ molecule$^{-1}$ s$^{-1}$ (Andersen et al., 2016). That the measured ambient temperature $k_1$ values determined in this work were indeed anomalously small is further confirmed by calculations using the most up-to-date structure activity relationship from Jenkin et al. (2018) which may be used to predict $k_1(298$ K) = $9.1 \times 10^{-12}$

cm$^3$ molecule$^{-1}$ s$^{-1}$. Even within the estimated factor-of-two accuracy of the SAR, the values of $k_1$ obtained from this work would appear anomalously small.

Results from the series of QCC detailed in section 3.3 above offer a potential explanation for the small $k_1(296$ K) values obtained in this work. As is common for reactions of oxygenated VOC, hydrogen bonded pre-reaction complexes (Fig. 9) were

key features of the most facile pathways to (R1) products (Fig. 10). However, the bulky methyl group substituents of TMO were found to hinder formation of the hydrogen-bonded complexes that would lead to abstraction from the more reactive CH$_2$ groups. Three reaction pathways remain open (Fig. 10), the most favourable proceeding via a H-bonded complex leading to abstraction from the CH$_3$ group, the other two being direct non-stabilised (no H-bonds) abstractions. The summative nature of the SAR has allowed for a simple modification to the calculations, whereby we have turned-off the (R1a) channel (Fig. 1)

identified by QCC as sterically hindered (section 3.3) and replaced this with abstraction from the equivalent CH$_2$ group in an unsubstituted aliphatic C$_5$ ring. This modified calculation allows us to use the SAR (Jenkin et al., 2018) to estimate a considerably smaller total $k_1(298$ K) = $3.8 \times 10^{-12}$ cm$^3$ molecule$^{-1}$ s$^{-1}$ for the remaining active channels: complexing followed by abstraction from the terminal CH$_3$ groups (R1b), and direct abstraction (no complexing) from the cyclic CH$_2$ groups. This modified SAR result is now within satisfactory agreement with the experimentally-determined value, given the estimated

factor-of-two accuracy of the SAR.



Some further evidence for the likely participation of hydrogen-bonded complexes was provided by the results obtained with PLP-LIF for $k_1$ over the range of temperatures 294 – 502 K (Fig. 6). Whilst the Arrhenius-like behaviour for $k_1(400 – 500$ K) is consistent with a simple direct abstraction mechanism, the "U-shaped" curve with a minimum at around 340 K is suggestive

of an increasingly important role for hydrogen-bonded pre-reaction complexes at lower temperatures. Experimental constraints meant that it was not possible to confirm these observations via the relative rate method, except at ambient temperature. However, confidence in the high-temperature PLP-LIF results may be derived from the good agreement between $k_{11}(T)$ obtained in this work (often "back-to-back" with $k_1(T)$ determinations) and an extensive literature dataset (Fig. 6). "U-shaped" Arrhenius plots have been observed for OH reactions with oxygenated VOC, notably acetone (Wollenhaupt et al., 2000) and

methyl pivalate (Wallington et al., 2001). In more recent work, experiments at low temperatures have revealed a dramatic increase in the rate of OH + CH$_3$OH (R16), with $k_{16}(63$ K) determined to be almost two orders of magnitude larger than $k_{16}(200$ K) (Shannon et al., 2013).

IUPAC have used a modified four-parameter Arrhenius-like expression Eq. (4) to represent $k(T)$ data for OH + acetone (Atkinson et al., 2006).

$k(T) = A_1\exp(-E_1 / T) + A_2\exp(-E_2 / T)$ Eq. (4)

Accordingly, the results from this work were fit with Eq. (4) to yield $k_1(294 – 502$ K) = $5.33\times10^{-10}$exp(-2237 / $T$) + $2.22\times10^{-13}$exp(766 / $T$). Also displayed on Fig. 6 are $k_1(294 – 370$ K) from the full SAR (Jenkin et al., 2018), exhibiting clear non-Arrhenius behaviour as these calculated values are dominated by the pathway forming complexes prior to abstraction at the reactive CH$_2$ groups. These calculated values clearly overestimate $k_1(294 – 370$ K). When the modification was applied

whereby this pathway was removed (see above), the modified SAR $k_1(294 – 370$ K), whilst not capturing the full complexity of $k_1(T)$, is now comfortably in agreement (within the SAR estimated accuracy of a factor of two) with our experimental determinations (Fig. 6).

Taken together, results from this work would indicate that, for atmospheric modelling purposes, OH + TMO proceeds largely

via abstraction from the terminal CH$_3$ groups (R1b) to generate the aldehyde product (Fig. 1). The best estimate available for an atmospheric removal rate coefficient is the room temperature value determined here of $k_1(296$ K) = $3.07 \times 10^{-12}$ cm$^3$ molecule$^{-1}$ s$^{-1}$. We note that $k_1$ does appear to be increasing at lower temperatures and that this value of $k_1(296$ K) may be an underestimate for cold conditions, e.g. the upper troposphere. Further $k_1$ and $k_2$ experiments at $T < 296$ K would be valuable but were not possible given the apparatus available in York and in Iasi.

**4 Atmospheric implications and conclusions**

An atmospheric lifetime ($\tau$) for TMO may be estimated using Eq. (5), based upon $k_1(296$ K) = $3.07 \times 10^{-12}$ cm$^3$ molecule$^{-1}$ s$^{-1}$ from this work and a value of [OH] = $1.13 \times 10^6$ molecule cm$^{-3}$ representative of the troposphere (Lelieveld et al., 2016).


$$\tau = \frac{1}{k_1[OH]}$$ Eq. (5)

The resultant $\tau_1 \approx 3$ days may be an overestimate for two reasons. First, as noted in section 3.4, our ambient-temperature value

may underestimate $k_1$ for colder troposphere conditions. Second, in regions highly impacted by atmospheric chlorine, the

lifetime of TMO may be constrained by (R2) with chlorine atoms. Estimates for ambient [Cl] vary widely and will anyway be

subject to a high degree of spatial variability. Nonetheless, using $k_2(296 \pm 2 \text{ K}) = (1.2 \pm 0.1) \times 10^{-10}$ cm$^3$ molecule$^{-1}$ s$^{-1}$ from

this work, and a large estimate of $[Cl] = 1 \times 10^4$ molecule cm$^{-3}$ (Li et al., 2018), we can demonstrate the feasibility of a lifetime

with respect to Cl atoms of $\tau_2 \approx 10$ days which, when combined with data for (R1), could reduce the overall atmospheric

lifetime for TMO. Losses of TMO to other atmospheric radicals, to $O_3$, or to photolysis appear unlikely and, to date,

unmeasured.

A lifetime calculation for $C_6H_5CH_3$, using (Eq. 5) with $k_3(298 \text{ K}) = 5.6 \times 10^{-12}$ cm$^3$ molecule$^{-1}$ s$^{-1}$ (Mellouki et al., 2021) and

$[OH] = 1.13 \times 10^6$ molecule cm$^{-3}$, yields $\tau_3 \approx 2$ days. It is clear, therefore, that TMO has a longer lifetime than the solvent it is

proposed to replace. A superficial assessment would indicate that, once emitted, TMO would have more time to disperse,

leading to a less spatially-concentrated build-up of harmful products such as $O_3$, HCHO, and other aldehydes.

In conclusion, the application of two distinctive experimental approaches together with QCC to the question of the atmospheric

fate of TMO has yielding the interesting result that TMO reacts with OH (R1) slower than predicted, with the consequence

being that TMO has a longer atmospheric lifetime than $C_6H_5CH_3$, the solvent it is proposed to replace. Further, the products

from (R1) are not the thermodynamic and kinetic (SAR) predicted products resulting from abstraction from the ring-methylene

(CH-H) group (R1a), but are more likely those resulting from abstraction from the terminal methyl (CH$_2$-H) groups (R1b).

This work has illustrated once again the importance of hydrogen-bonded pre-reaction complexes for OH + oxygenated VOC

reactions and the need for combined lab and computational studies to properly understand such reaction mechanisms.

**Author contribution**

Laser-based experiments were designed by TJD and conducted by JVS, CM, AH and TJD. Chamber experiments were

designed by IGB, CA and RIO and conducted by CM, CR and IGB. Quantum chemical calculations were carried out by JVS

and CDR. The manuscript was written by CM, JVS, CDR and TJD with assistance from other authors. TJD, FB and CRM

conceived of the overall project.

**Acknowledgements**

The chamber experiments from this work received funding from the European Union's Horizon 2020 research and innovation

programme through the EUROCHAMP-2020 Infrastructure Activity under grant agreement No 730997. JVS acknowledges



support from the EU ERASMUS programme, CM thanks the Dept. of Chemistry at York for a PhD scholarship. CR, CA, RIO and IGB acknowledge support from PN-III-P4-PCE2021-0673 UEFISCDI project. The authors thank the York support team

and in particular Danny Shaw, Abby Mortimer, Mark Roper, Stuart Murray and Chris Rhodes for always excellent technical support. Luc Vereecken and Andrew Rickard are thanked for helpful discussions regarding the reactivity of TMO. Finally, the authors thank Katherine Manfred for the precious help with PLP-LIF and Labview.





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





**Figure 1.** The chemical structure of TMO and the two possible breakdown routes following reaction (R1) with OH. The differing reactivities of the ketone (R1a) and aldehyde (R1b) products with respect to reaction with OH and to photolysis will control subsequent breakdown chemistry and ultimately the impact of TMO on air quality metrics such as $O_3$ production. Note that reaction of TMO with Cl (R2) would proceed via two equivalent pathways.




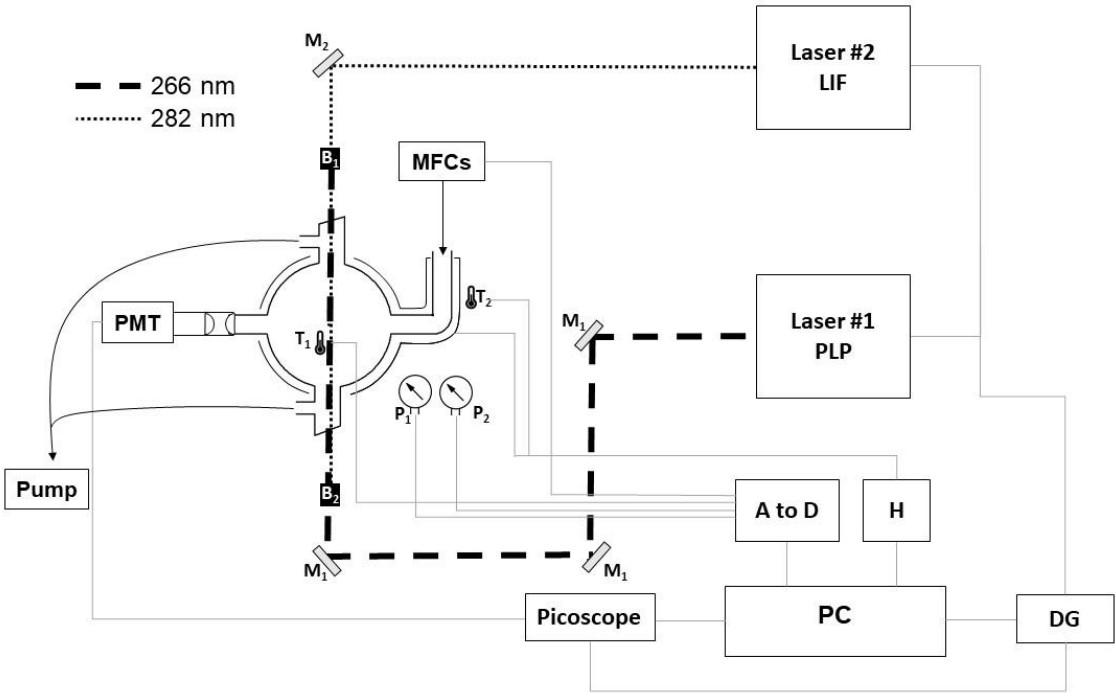


**Figure 2.** Schematic of the PLP-LIF apparatus. Key: Laser#1 = 266 nm YAG for PLP OH generation; Laser#2 = 532 nm YAG-pumped dye, frequency doubled for output at 282 nm for OH LIF detection; M1 = 266 nm dichroic mirrors; M2 = 282 nm dichroic mirror; B1 and B2 = beam stoppers; T1 and T2 = thermocouples; P1 and P2 = 1000 Torr and 10 Torr pressure gauges; DG = delay generator; H = thermal control box; A to D = analogue to digital converter to allow PC control of gas flows and logging of cell temperature and pressure.






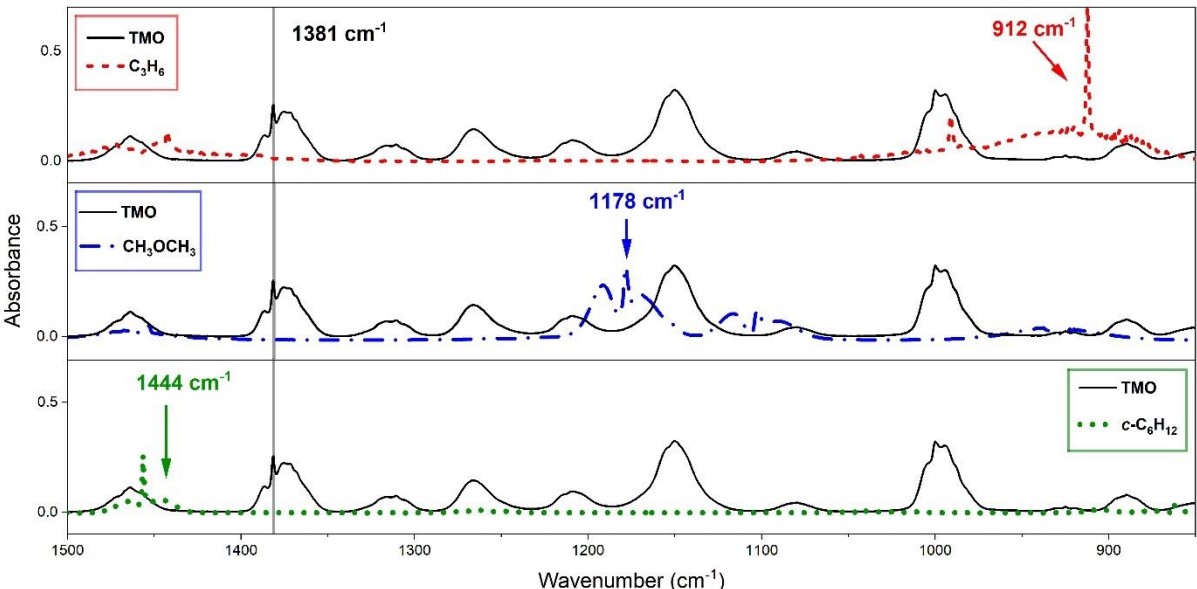

**Figure 3.** Observed FTIR spectra from this work with typical monitoring frequencies indicated: TMO (black solid line), monitored at 1381 cm$^{-1}$ where reference compounds did not appreciably absorb; for C$_3$H$_6$ (the red dashed line) at 1381 cm$^{-1}$; CH$_3$OCH$_3$ (blue dot-dash line) and $c$-C$_6$H$_{12}$ (green dotted line) at 1444 cm$^{-1}$.

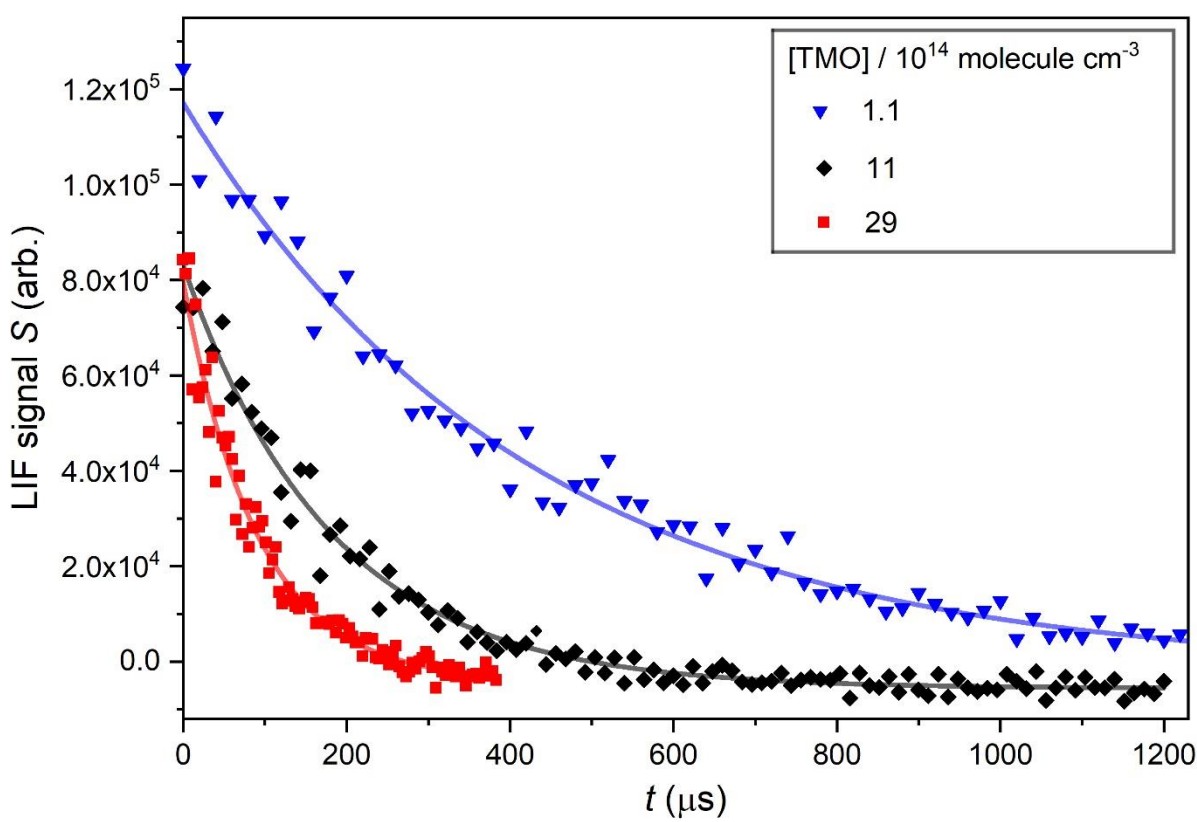

**Figure 4.** Displays OH decays obtained in PLP-LIF experiments using three different [TMO], each fit with Eq. (1) to determine pseudo first-order rate coefficients: $B = (2390 \pm 69)$ s$^{-1}$ at [TMO] = 1.1×10$^{14}$ molecule cm$^{-3}$; $B = (5530 \pm 172)$ s$^{-1}$ at [TMO] = 11×10$^{14}$ molecule cm$^{-3}$; and $B = (11200 \pm 464)$ s$^{-1}$ at [TMO] = 29×10$^{14}$ molecule cm$^{-3}$.



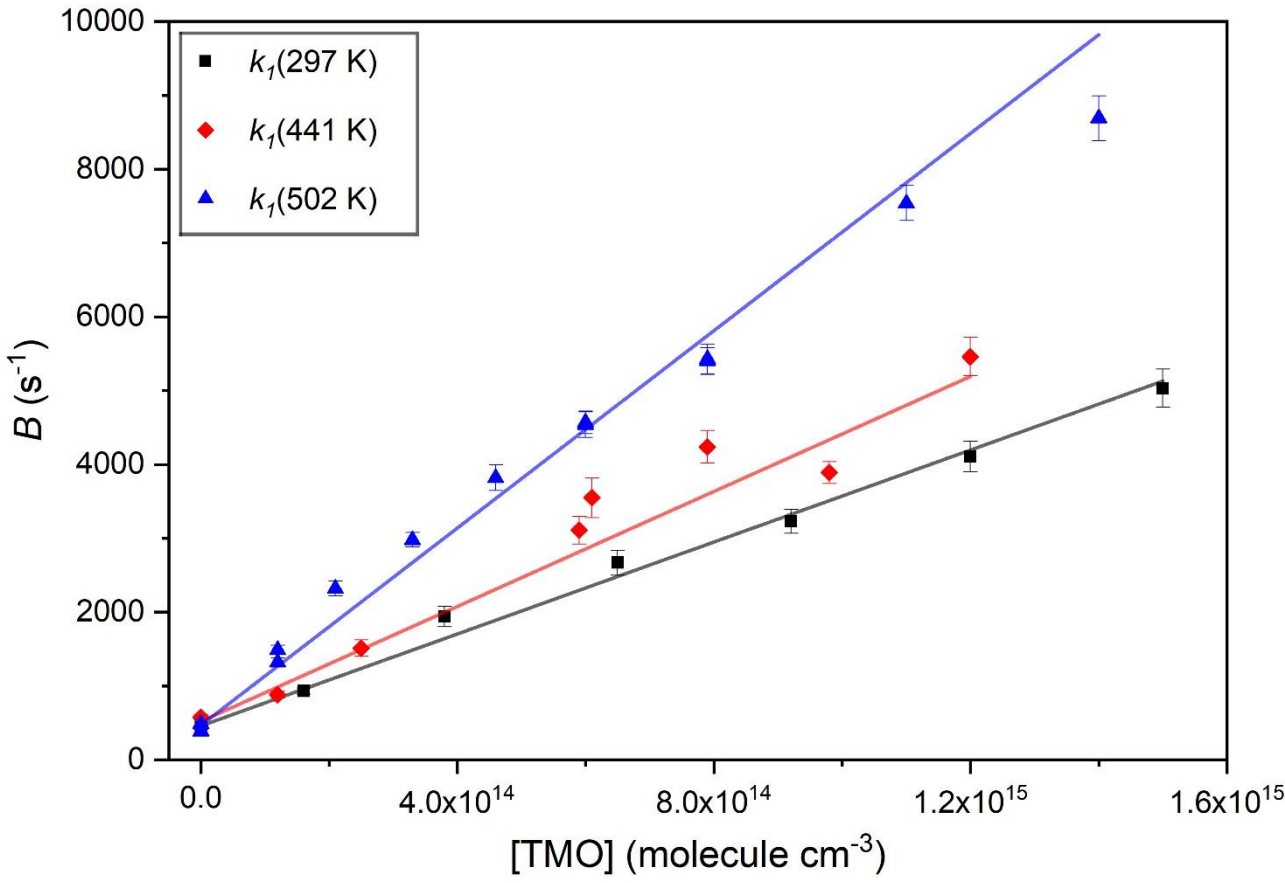

**Figure 5.** Three typical bimolecular plots of $B$ vs. [TMO] used to obtain $k_1(T)$ for OH + TMO, each fit with Eq. 2 to obtain: $k_1(297\ \mathrm{K}) = (3.1 \pm 0.1) \times 10^{-12}\ \mathrm{cm^3\ molecule^{-1}\ s^{-1}}$, from the blue triangles; $k_1(441\ \mathrm{K}) = (3.6 \pm 0.3) \times 10^{-12}\ \mathrm{cm^3\ molecule^{-1}\ s^{-1}}$, from the red diamonds; $k_1(502\ \mathrm{K}) = (5.9 \pm 0.3) \times 10^{-12}\ \mathrm{cm^3\ molecule^{-1}\ s^{-1}}$, from the black squares.




**Figure 6.** Arrhenius plot displaying $k_1(T)$ results from this work obtained via PLP-LIF (filled blue diamonds) and RR (filled blue triangle) all fit with the four-parameter Eq. (4). to yield $k_1(294 - 502 \text{ K}) = 5.33 \times 10^{-10} \exp(-2237 / T) + 2.22 \times 10^{-13} \exp(+766 / T)$ cm$^3$ molecule$^{-1}$ s$^{-1}$ (the blue dashed line). Calculated $k_1(298 - 370 \text{ K})$ using the SAR proposed by Jenkin et al. (2018) is displayed as the blue dotted line whilst the blue dot-dashed line represents a modified SAR calculation (see section 3.4). Also displayed are $k_{16}(299 - 501 \text{ K})$ from this work (filled red squares) alongside literature determinations (Hess and Tully, 1989; Wallington et al., 1988; Jiménez et al., 2003; Dillon et al., 2005) and the most recent evaluation from IUPAC (Atkinson et al., 2006), recommending $k_{16}(210 - 866 \text{ K}) = 6.38 \times 10^{-18} T^2 \exp(144 / T)$ cm$^3$ molecule$^{-1}$ s$^{-1}$ (the solid red line).





**Figure 7.** Exemplary relative rate plots, used to determine $k_1(296\ K)$ for OH + TMO (R1). The red circles correspond to use of $CH_3OCH_3$ (R10) as reference; the black squares $c\text{-}C_6H_{12}$ (R11). The solid lines are linear fits, with gradient values used (Eq.3) in conjunction with literature data (Table 2) to obtain $k_1$. Each experiment was repeated twice, and the full results, summarised in Table 4, were averaged to obtain $k_1(296\ K) = (3.07 \pm 0.05) \times 10^{-12}\ cm^3\ molecule^{-1}\ s^{-1}$.



**Figure 8.** Exemplary relative rate plots used to determine $k_2$(296 K) for Cl + TMO (R2). Gradients obtained from linear fits (the solid lines) were used in conjunction with literature data (Table 2) to obtain $k_2$. Each experiment was repeated twice, and the full results, summarised in Table 5, were averaged to obtain $k_2$(296 K) = (1.2 ± 0.1) × 10$^{-10}$ cm$^3$ molecule$^{-1}$ s$^{-1}$.





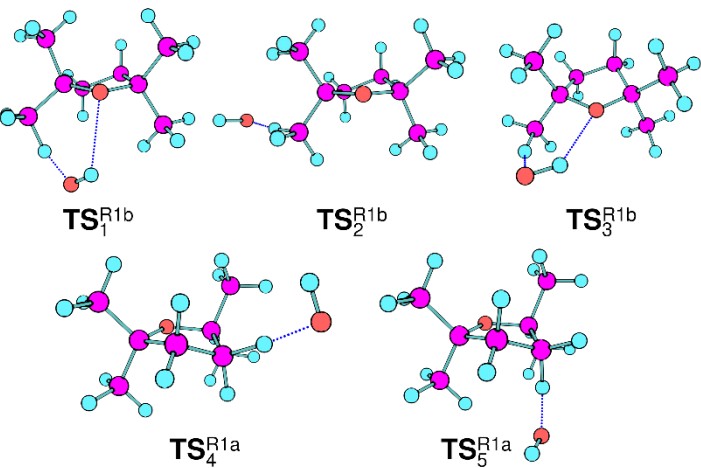

**Figure 9.** The five transition states (TS) characterised via QCC (section 3.3) of which $TS_2^{R1b}$, $TS_4^{R1a}$, and $TS_5^{R1a}$ are associated with 'direct', unstabilised hydrogen abstraction; $TS_1^{R1b}$ and $TS_3^{R1b}$ are associated with stabilised hydrogen abstraction that proceeds via a hydrogen-bound pre-reaction complex and TS. See Fig. 1 for description of reaction channels proceeding to (R1a) and (R1b) products.





**Figure 10.** Free energy profile of (R1) hydrogen abstraction pathways: two leading to (R1a) products via ($TS_{4-5}^{R1a}$) and three ($TS_{1-3}^{R1b}$) leading to (R1b) products, labelled according to the associated transition state in the legend. For each of the five pathways along the reaction coordinate, five stationary point energies (1 = separated reactants, 2 = pre-reaction complex, 3 = transition state, 4 = post-reaction complex and 5 = separated products) were calculated using G4 model chemistry, $p = 1$ bar and $T = 298$ K. Several were degenerate or near-degenerate, notably the pre-reaction complexes leading to TS1 and TS3 (see Table 6 for details).