# Peer review of "Atmospheric breakdown chemistry of the new green solvent 2,2,5,5tetramethyloxolane via gas-phase reactions with OH and Cl radicals"

_Atmospheric Chemistry and Physics, 2022_

## Referee Comment (RC2)

**Atmospheric breakdown chemistry of the new green solvent 2,2,5,5- tetramethyloxolane via gasphase reactions with OH and Cl radicals**

**Overall**, general comment**

This study has determined the gas-phase rate coefficients for the removal of tetramethyloxolane (TMO) with either hydroxyl (the main atmospheric oxidant) or chlorine radicals. Direct monitoring of OH in the presence TMO and TMO removal compared to another reagent – relative rate - provided two complimentary methods for determining the OH + TMO rate coefficient at room temperature. Excellent agreement was observed. The direct monitoring experiments were able to study the reaction over a range of elevated temperatures, up to 502 K, and the relative rate method also determined the Cl + TMO rate coefficient at room temperature. This experimental work has been carried out with enough suitable checks to give good confidence in their results. *Ab initio* calculations have been performed to indicate the site of H-abstraction, mainly at the CH3 is suggested. Based on these experiments, I have no problem recommending this paper for publication. But the presentation of the calculations should be improved: show the potential energy surface and give the vibrational frequencies of the species in the SI.

**Specific comments**

To add further interpretation to the OH + TMO experimental results *ab initio* calculations have been carried out, where H-abstraction from either the CH3 or the CH2-ring was investigated. It is suggested that H-abstraction from the CH3 is dominant at room temperature. In principle, enough information from these calculations is present to then carry out reaction rate theory calculations but this has not been done. Such calculations would better highlight the branching between CH3 and CH2 abstraction. It is conventional when doing such calculations to present the potential energy surface, but instead the free energy surface at 298 K has been presented. Also, no vibrations of the identified species of the reaction have been given in the SI. I recommend this be given in the supporting information.

In the discussion, it is highlighted that the observed OH + TMO rate coefficient is significantly smaller than predicted from the literature structural activity relationship, SAR, which is based on H-abstraction from the CH2. Recognizing that abstraction is mainly via H-abstraction at the CH3 allows a modified SAR to be constructed that is a better representation of the data, but still far from good. It is well-know that SAR struggle to fully describe experimental data over a range of temperatures when the reaction involves H-bonded complexes. As well as the *ab initio* calculations, further evidence of the involvement of H-bonded complexes in the reaction is supported by the "U-shaped" curved in the kinetic data, where at 340 K the minimum rate coefficient is observed. Based on Figure 6, the SAR and the modified SAR are straight-lines, but the experimental data is "curved". Is it possible for a SAR to be "curved"?

As noted by RC1, is it reasonable to claim that "TMO is less problematic VOC than toluene"? Is this statement based on its rate coefficient alone? If this is the case then in an environment where Cl initiated oxidation dominants, then it is worse than toluene, as Cl + toluene is  $5.8 \times 10^{-11}$ ! More of the full oxidation pathway needs to be taken into account. While it is suitably known for toluene, is it known for TMO beyond Figure 1?

**Line by Line: technical**

16 "computational experiments"?

49 "with a photochemical lifetime of around two days" I presume you mean wrt to OH, R3?

**57** "In air heavily impacted by atmospheric chlorine (Ariya et al., 1999; Atkinson and Aschmann, 1985; Thornton et al., 2010), TMO loss may be augmented by reaction with Cl atoms (R2)." Are the likely sites of toluene / TMO emissions going to coincide with areas where [Cl] are significant?

**107** *"situated downstream from a MFC and thus operating at close to reactor pressure"* How do you estimate how much this adds to the total flow through the system? You need to know this.

**174** " $S(t) = S_0 exp(-Bt)$ " Was there a baseline to worry about. Sometimes this is a probe induced problem, and sometimes there is an offset in the signal because of your measurement device.

**178** " $[H_2O_2] \cdot 1 \cdot 10_{14}$  molecule  $cm_{-3}$ " Is this concentration based on the intercept in Fig. 4? Using this and a typical photolysis energy allows an estimate of [OH]. My estimate is <=  $10^{12}$  molecule cm3.

**199** "A mean of four values obtained at around room temperature yielded  $k1(296 \text{ K}) = (3.07 \pm 0.04) \times 10-12 \text{ cm3}$  molecule-1 s-1" If I look at Table 3, the 4 reported errors in the rate coefficients determination are close to 10%. So it is not clear how you have taken this data and assigned close to 1% error. Were the errors in Table 1 used? The uncertainty in flows and reagent concentrations normally means the accuracy of the measurement is higher than the quoted error.

**236** "A weighted mean value from these four relative rate determinations was  $k1,RR(296 \text{ K}) = (3.07 \pm 0.05) \times 10 - 12 \text{ cm3}$ " As noted above, weighted mean seems to lead to unrealistically small errors. If I look and Figures 7 and 8, the data in Figure 8 appear to have less systematically variation but has an error in k2 of ~ 8%, while the k1 has a quoted error of ~2%!

Are the errors quotes in this paper 1 or 2 sigma? Later in the discussion the above points are addressed by stating the overall uncertainty as just over 10%. This seems reasonable.

**252** "explore the hydrogen abstraction pathways (R1a) and (R1b) set out in Figure 1 and thus predict the products of (R1)." To be clear, the *ab initio* calculations have only explored the abstraction. The potential products from these radicals are based on comparison other reactions? Was there evidence of the formation of these carbonyl products in the relative rate studies?

**261** *"The free energy profiles of the reaction channels are presented in Figure 10"* These calculations could be presented in a more informative way. There should be enough information from these calculations to then do rate theory calculations, but was not attempted here. In order for someone else to make use of this data they would need the vibrations of all the species. This information is not given; please add it to the SI; also give the rotational constants. It is more conventional to show potential energy surfaces, PES, i.e. the energy of the reaction, enthalpy at zero K, rather than the free energy. By presenting free energy you are omitting giving the vibrational frequencies of the species.

Rate theory is statistical and only considers the turning points on the PES, so the distance between the species is not particularly that important. So, I do not consider the distance between the TS and pre-reaction complexes and *r* that important:

**262** mass-weighted distances (MWDs) quantifying the separation in Cartesian space between the TS and pre-reaction complexes, and key structural parameters (rO–H, rO···H, and aH–O···H) are tabulated for each of the TS in Table 6.

If the PES was shown, then I suspect there would be a strongly bound Pre-RC - ca. 20 kJ mol-1 - and less strongly bound Pre-RC. And at low*T*, it will probably proceed mainly via strongly bound Pre-RC. But without rate theory this is conjecture.

**278** "The intimation is that the hydrogen-bound pre-reaction complex acts as a funnel on the free energy surface to bias the reaction towards preferential production of the kinetic (via R1b)" Yes, this will be generally the case at low *T*, but as *T* is increased the influence of the complex will be lost and the reaction will be controlled by TS. I think this is what your free energy is suggesting. If you calculated the free energy at say 200, 300, 400 and 500 K, it might show that the influence of the pre-RC is lost and the reaction rates are controlled almost solely by TS. I suspect, the fact that the rate coefficient start to increase above 350 K is where the influence of the complex is lost.

**P313** *"the other two being direct non-stabilised (no H-bonds) abstractions."* In Fig10 and SI, Pre-RC are given for all reactions? If there is direct abstraction, is there a Pre-RC?

The claim in the abstract that TMO is a less problematic solvent than toluene and **364**:

It is clear, therefore, that TMO has a longer lifetime than the solvent it is proposed to replace. A superficial assessment would indicate that, once emitted, TMO would have more time to disperse, leading to a less spatially-concentrated build-up of harmful products such as O3, HCHO, and other aldehydes.

Should be further qualified. It is just based on its slower OH + TMO rate coefficient ("superficial") rather than considering the full oxidation pathway, which I presume is not so well known for TMO compared to toluene.

---

## Author Response (AR1)

**Author Response to Referee Comments**

We thank both reviewers for their time and attention and for the insightful and constructive comments on our research paper. The suggestions have all contributed to an improved manuscript.

**Response to RC1:**

Referee comments in *italic blue*; our response in **bold black.**

This paper reports the results of a careful experimental and computational study of the kinetics of the reactions of hydroxyl radicals and chlorine atoms with a new solvent, 2,2,5,5-tetramethyloxolane. The results are presented in a clear and logical fashion and I recommend publication essentially as is.

I have only one suggestion for improvement which is to remove the claim of "green" for the subject molecule in the title (which sounds a little unscientific) and to add a quantitative justification for the conclusion in the abstract that TMO is a "less problematic" VOC than toluene. Jenkin et al. (Atmos. Environ., 163, 128, 2017) have provided a method to estimate the photochemical ozone creation potential (POCP) from molecular structure and k(OH). This method could be used to estimate the POCP for TMO which could then be compared to that for toluene (and other solvents). Citation: https://doi.org/10.5194/acp-2022-446-RC1

We agree that quantification of the air quality impact of TMO relatively to toluene is useful. Many thanks for the suggestion. Using the method from Jenkin et al. (2017), we have estimated the photochemical ozone creation potential (POCPE) for TMO to be 18 in NW Europe conditions. Toluene has a considerably larger POCPE value of 45. This information is now included in the Atmospheric Implications section of our revised manuscript, together with details of these calculations in supplemental information.

Regarding the title and description of TMO as a "new green solvent", we have put the word green in quotation marks. We could think of no suitable, succinct phrase to describe a solvent such as TMO that is reportedly sustainably-sourced, bio-derived, non-toxic and non-hazardous. Scientists at University of York applied the well-established twelve principles of Green Chemistry when designing and developing TMO (Anastas and Eghbali, 2010; Byrne et al., 2017).

**Response to RC2:**

Referee comments in *blue italic;* our response in **bold black**.

This study has determined the gas-phase rate coefficients for the removal of tetramethyloxolane (TMO) with either hydroxyl (the main atmospheric oxidant) or chlorine radicals. Direct monitoring of OH in the presence TMO and TMO removal compared to another reagent – relative rate – provided two complimentary methods for determining the OH + TMO rate coefficient at room temperature. Excellent agreement was observed. The direct monitoring experiments were able to study the reaction over a range of elevated temperatures, up to 502 K, and the relative rate method also determined the Cl + TMO rate coefficient at room temperature. This experimental work has been carried out with enough suitable checks to give good confidence in their results. Ab initio calculations have been performed to indicate the site of H-abstraction, mainly at the CH3 is suggested. Based on these experiments, I have no problem recommending this paper for publication. But the presentation of the calculations should be improved: show the potential energy surface and give the vibrational frequencies of the species in the SI.

Many thanks for the helpful commentary. The presentation of the calculations has been improved and the PES and vibrational frequencies are presented in the revised manuscript and Supporting Information.

**Technical comments:**

- **16** "*computational experiments*"? This line in the abstract now reads "The atmospheric chemistry of 2,2,5,5-tetramethyloxolane (TMO), a promising "green" solvent replacement for toluene, was investigated in laboratory-based experiments and computational calculations."
- 49 "with a photochemical lifetime of around two days" I presume you mean wrt to OH, R3? Yes. Now clarified as "IUPAC have evaluated the kinetic literature for (R3) and recommend a rate coefficient, k3(298 K) = 5.6×10-12 cm3 molecule-1 s-1 with an associated uncertainty Δlogk3(298 K) = 0.10 or approximately 25% (Mellouki et al., 2021)"
- 57 "In air heavily impacted by atmospheric chlorine (Ariya et al., 1999; Atkinson and Aschmann, 1985; Thornton et al., 2010), TMO loss may be augmented by reaction with Cl atoms (R2)." Are the likely sites of toluene / TMO emissions going to coincide with areas where [Cl] are significant? This will vary from site to site. Since industrial centres are often located in coastal areas, around estuaries and ports (Lotze et al., 2006), we do think Cl may play a role in TMO chemical degradation in some scenarios. We are happy with the existing introductory text on this issue.
- -
  - **107** "situated downstream from a MFC and thus operating at close to reactor pressure" How do you estimate how much this adds to the total flow through the system? You need to know this. We do need to know that any additional flow from the bubbler is insignificant compared to the total flow. This was indeed the case (<< 1% overall flow), as verified in three ways: 1) no change in reactor pressure observed upon opening the H2O2 bubbler; 2) by considering the vapour pressure of the H2O2 / H2O mixture (100 Pa for 80% H2O2, www.h2o2.com/technical-library/physical-chemical-properties/physical-properties/default.aspx?pid=25&name=Vapor-Pressures) which was too small to contribute significantly; 3) from our (kinetic) measurements of [H2O2]  $\approx 10^{14}$  molecule cm-3 which (whilst neglecting the contribution of H2O) demonstrate that mass transfer from the bubbler was insignificant compared to the overall gas concentration of 1018 molecule cm-3 in these experiments.

- 174 "S(t) = S0 exp(-Bt)" Was there a baseline to worry about. Sometimes this is a probe induced problem, and sometimes there is an offset in the signal because of your measurement device. Indeed. Line 174 now reads "PLP-LIF studies were carried under pseudo-first order conditions of [TMO] >> [OH] such that (following subtraction of measured baseline) OH LIF time profiles, S(t), were described by a monoexponential decay, expression Eq. (1):"
- **178** "[H2O2] 1 1014 molecule cm-3" Is this concentration based on the intercept in Fig. 4? Using this and a typical photolysis energy allows an estimate of [OH]. My estimate is  $\langle = 10^{12} \text{ molecule cm}^3$ . The calculation was indeed based on intercept values from Fig. 4 but also used an erroneous value for the laser fluence of 100 mJ / pulse. This was the manufacturer specification fluence, whereas upon measuring up- and down-stream of the reactor we were able to report a more realistic value of 20 mJ / pulse. The manuscript has been corrected, with line 88 reading "E = 20 mJ / pulse" and line 96 "[OH]  $\approx$  5x1011 molecule cm-3".
  - **199** "A mean of four values obtained at around room temperature yielded  $k1(296 \text{ K}) = (3.07 \pm 0.04) \times 10^{-12} \text{ cm}^3$  molecule-1 s-1" If I look at Table 3, the 4 reported errors in the rate coefficients determination are close to 10%. So it is not clear how you have taken this data and assigned close to 1% error. Were the errors in Table 1 used? The uncertainty in flows and reagent concentrations normally means the accuracy of the measurement is higher than the quoted error. The value k1(296 K) = (3.07 \pm 0.04) \times 10^{-12} \text{ cm}^3 molecule-1 s-1 is the calculated mean value, weighted to the error bars of individual measurements. Smaller standard errors were the result of these repeated measurements. The referee is correct to point out that uncertainties in flows and hence reagent concentrations are significant; to account for these we quoted a more realistic value of k1(296 K) = (3.1 \pm 0.4) \times 10^{-12} \text{ cm}^3 molecule-1 s-1 in the discussion, atmospheric implications and abstract.
- **236** "A weighted mean value from these four relative rate determinations was  $k1, RR(296 \text{ K}) = (3.07 \pm 0.05) \times 10^{-12} \text{ cm}^3$ " As noted above, weighted mean seems to lead to unrealistically small errors. If I look and Figures 7 and 8, the data in Figure 8 appear to have less systematically variation but has an error in k2 of ~ 8%, while the k1 has a quoted error of ~2%! Are the errors quotes in this paper 1 or 2 sigma? Later in the discussion the above points are addressed by stating the overall uncertainty as just over 10%. This seems reasonable. As presented in Tables 4 and 5, all relative-rate datasets have statistical errors (one sigma) of between 2% and 8%. The data presented in Figures 7 and 8 are consistent with this statement. The (R1) datasets presented in Fig 7 were chosen for clarity of presentation (data from one experiment not obscuring the other) but are not the least noisy.
- 252 "explore the hydrogen abstraction pathways (R1a) and (R1b) set out in Figure 1 and thus predict the products of (R1)." To be clear, the ab initio calculations have only explored the abstraction. The potential products from these radicals are based on comparison other reactions? Was there evidence of the formation of these carbonyl products in the relative rate studies? Indeed, the QCC only explore abstraction pathways but in so doing, do allow for predictions of e.g. ketone and aldehyde product yields. Line 252 has been corrected to read "QCC were carried out at the CBS-QB3 (Montgomery et al., 1999) and G4 (Curtiss et al., 2007) model chemical levels of theory (Section 2.3) to investigate the small k1(296 K) and complex k1(T) determined in laboratory studies, explore the hydrogen abstraction pathways (R1a) and (R1b) set out in Figure 1 and thus enable prediction of product yields. No evidence of first generation product formation was observed in the FTIR experiments; this was likely due to probably higher reactivity towards OH of these aldehyde and ketone products.

- **261** "The free energy profiles of the reaction channels are presented in Figure 10." These calculations could be presented in a more informative way. There should be enough information from these calculations to then do rate theory calculations but was not attempted here.
- ... "In order for someone else to make use of this data they would need the vibrations of all the species. This information is not given; please add it to the SI; also give the rotational constants." The vibrational frequencies at the B3LYP/CBSB7 (CBS-QB3) and B3LYP/GTBas3 (G4) levels of theory for all structures have been added to the expanded SI in Tables S41-80. The rotational constants have also been added at the B3LYP/CBSB7 (CBS-QB3) and B3LYP/GTBas3 (G4) levels of theory for all structures in Tables 81-120.
- ... "It is more conventional to show potential energy surfaces, PES, i.e. the energy of the reaction, enthalpy at zero K, rather than the free energy... if the PES was shown, then I suspect there would be a strongly bound Pre-RC ca. 20 kJ mol-1 and less strongly bound Pre-RC." Figure 10b in the SI has been added to reflect the potential energy, rather than free energy, surface. This is observed, as predicted.
- 278 "The intimation is that the hydrogen-bound pre-reaction complex acts as a funnel on the free energy surface to bias the reaction towards preferential production of the kinetic (via R1b)" Yes, this will be generally the case at low T, but as T is increased the influence of the complex will be lost and the reaction will be controlled by TS. I think this is what your free energy is suggesting. If you calculated the free energy at say 200, 300, 400 and 500 K, it might show that the influence of the pre-RC is lost and the reaction rates are controlled almost solely by TS. I suspect, the fact that the rate coefficient start to increase above 350 K is where the influence of the complex is lost. The Reviewer is exactly right; the influence of the pre-RC is lost with increasing temperature. The Figure below shows the free energy profile of the reaction channel through  $TS_1^{R1b}$  (the channel with the most stable H-bound pre-RC) at a range of temperatures between 200 and 500 K. The influence of the pre-RC is indeed lost around 350-400 K, exactly as the Reviewer predicted, coinciding with the increase of the rate coefficient. We have added a sentence on this to the revised manuscript, with line 282 reading: "As the temperature is increased, the influence of the hydrogenbound pre-reaction complex is commensurately reduced and the reaction is controlled increasingly by  $\Delta G^{\sharp}$ . The OCC predict that control passes over to  $\Delta G^{\sharp}$  at ca. 350-400 K (at which point the hydrogen-bound pre-reaction complex is no longer stabilised relative to the reactants), consistent with the temperature at which k(T) is observed to increase in our experimental data."

Free Energy Reaction Profile | B3LYP/CBSB7

**Figure.** Relative free energy ( $\Delta G$ ) profile for the reaction channel through  $\mathbf{TS}_1^{\text{R1b}}$  at a range of temperatures, *T*, between 200 and 500 K. *p* = 1 bar. B3LYP/CBSB7.

- 313 "the other two being direct non-stabilised (no H-bonds) abstractions." In Fig10 and SI, Pre-RC are given for all reactions? If there is direct abstraction, is there a Pre-RC?" There are not any "stabilised" pre-RCs (the pre-RCs located are in higher-energy regions of the potential energy surface along the TMO approach channels for OH and are unstable with respect to collapse, regenerating the reactants) but, nonetheless, loosely-bound structures with O…H interactions can be converged successfully and obtained as real minima. The O…H interaction distance in these pre-RCs is approximately 1.8-2.0 Å [comparable to the interaction distance (1.8 Å) in the stabilised, H-bound pre-RC] and the effect of the weak O…H interaction is reflected in the slightly elongated (+0.1 Å) C-H bonds *cf*. TMO in these structures.

Anastas, P. and Eghbali, N.: Green Chemistry: Principles and Practice, Chem. Soc. Rev., 39, 301-312, 10.1039/B918763B, 2010.

Byrne, F., Forier, B., Bossaert, G., Hoebers, C., Farmer, T. J., Clark, J. H., and Hunt, A. J.: 2,2,5,5-Tetramethyltetrahydrofuran (TMTHF): a non-polar, non-peroxide forming ether replacement for hazardous hydrocarbon solvents, Green Chem., 19, 3671-3678, 10.1039/C7GC01392B, 2017.

Lotze, H. K., Lenihan, H. S., Bourque, B. J., Bradbury, R. H., Cooke, R. G., Kay, M. C., Kidwell, S. M., Kirby, M. X., Peterson, C. H., and Jackson, J. B. C.: Depletion, degradation, and recovery potential of estuaries and coastal seas, Science, 312, 1806-1809, 10.1126/science.1128035, 2006.

Mellouki, A., Ammann, M., Cox, R. A., Crowley, J. N., Herrmann, H., Jenkin, M. E., McNeill, V. F., Troe, J., and Wallington, T. J.: Evaluated kinetic and photochemical data for atmospheric chemistry: volume VIII - gas-phase reactions of organic species with four, or more, carbon atoms (>= C-4), Atmos. Chem. Phys., 21, 4797-4808, 10.5194/acp-21-4797-2021, 2021.